# Confirmatory Factor Analysis of the Malay Version of the Smartphone Addiction Scale among Medical Students in Malaysia

**DOI:** 10.3390/ijerph17113820

**Published:** 2020-05-28

**Authors:** Siti Rubiaehtul Hassim, Wan Nor Arifin, Yee Cheng Kueh, Nor Azwany Yaacob

**Affiliations:** 1Biostatistics and Research Methodology Unit, School of Medical Sciences, Universiti Sains Malaysia, Kubang Kerian 16150, Malaysia; bisyarah_istikharah@yahoo.com (S.R.H.); yckueh@usm.my (Y.C.K.); 2Department of Community Medicine, School of Medical Sciences, Universiti Sains Malaysia, Kubang Kerian 16150, Malaysia; azwany@usm.my

**Keywords:** confirmatory factor analysis, Malay version, medical student, smartphone addiction, validation study

## Abstract

Background: At present, the validity and reliability evidence of the Malay version of the Smartphone Addiction Scale (SAS-M) is only available by exploratory factor analysis (EFA). The aim of this study is to validate and determine the psychometric properties of the SAS-M by confirmatory factor analysis (CFA). Methods: A cross-sectional study was conducted among 323 medical students in Universiti Sains Malaysia. The students were given questionnaire forms consisting of socio-demographic information, the SAS-M and the Malay version of the Internet Addiction Test (MVIAT). The CFA was conducted using robust maximum likelihood estimator. The internal consistency reliability was determined by Raykov’s rho coefficient. The concurrent validity was assessed by the Pearson’s correlations between the factor scores of the SAS-M and the MVIAT. Results: The analysis showed the five-factor model of the SAS-M has an acceptable model fit after the inclusion of 12 correlated errors (SRMR = 0.067, RMSEA 0.059 (90% CI: 0.054, 0.065), CFI = 0.895, TLI = 0.882). The factor loadings ranged from 0.320 to 0.875. The internal consistency reliability was good (Raykov’s rho = 0.713 to 0.858) and it showed good concurrent validity with the MVIAT. Conclusions: The CFA showed that the SAS-M is a valid and reliable self-administered questionnaire to measure the level of smartphone addiction among medical students.

## 1. Introduction

Nowadays, a smartphone is an essential and popular communication tool that makes our daily life more convenient. As compared to a phone with basic features, a smartphone packs more computing capability and connectivity [1]. A smartphone performs many of the functions of a computer and comes with an operating system that allows it to run mobile applications [2,3]. The hardware capability, connectivity and inclusion of operating systems allow a smartphone to have a wide range of software, Internet and multimedia (music, video, camera and gaming) functionality, alongside main phone functions such as voice calls and text messaging [3]. All these functions come in the form of a pocket-sized device and its popularity is inevitable [4]. Widespread and pervasive smartphone use has become the social norm because of the high accessibility and mobility of the device [5].

In Malaysia, the smartphone penetration rate stands at a staggering 97.9% based on the latest report by the Department of Statistics, Malaysia in 2019 [6]. The smartphone has become the most common device to access the Internet based on the survey by the Malaysian Communications and Multimedia Commission (MCMC) in 2018, which reported 93.1% of Internet users used smartphones to go online [7]. Based on another survey by MCMC in 2017, other common uses of smartphones are text and voice messaging (98.5%), voice calls (93.8%), social media (88.1%), entertainment (83.7%), map/navigation (73.6%), email (60.0%) and video calls (53.2%) [8]. Judging from these nationwide surveys, the findings clearly show that the smartphone permeates the life of all Malaysians.

There is an increasing concern about the negative effects that come from its overuse, and these may interfere with our daily life [4]. De-Sola Gutiérrez et al. [9] define smartphone addiction as an excessive attention and uncontrolled dedication to one’s smartphone. It is a behavior that manifests tolerance, withdrawal symptoms and dependence, along with social problems [10]. Studies on the effects of smartphone use on students have revealed many negative sides of smartphone addiction [11]. Smartphone addiction is associated with low academic performance [12,13], poor attention during learning sessions [12,14], poor psychological health [15,16], poor social interaction [17,18,19] and low life satisfaction [13].

Studies have also attributed problematic smartphone use with the social media use of smartphones [12,20].

A number of local studies have also supported the negative effects of smartphone use among students in Malaysia. In a study among undergraduate students, Ithnain, Ghazali and Jaafar [21] concluded that smartphone addiction is associated with depression and anxiety. In another study among medical students, Hadi et al. [22] found the association of smartphone addiction with poor psychological health. A recent study reported that 60.7% of undergraduate students had problematic smartphone use, and this was found to be associated with anxiety, depression, stress and a deterioration in academic performance [23]. An earlier study also supported the relationship between problematic phone use with lower academic performance [24].

Given the importance of assessing smartphone-related problems, so far nine major scales have been used in studies, which are the Smartphone Addiction Scale (SAS), Smartphone Addiction Scale Short Version (SAS-SV), Smartphone Addiction Proneness Scale (SAPS), Mobile Phone Addiction Index (MPAI), Mobile Phone Addiction Scale (MPAS), Mobile Phone Problem Use Scale (MPPUS), Smartphone Addiction Scale (SPAS), Smartphone Addiction Inventory (SPAI), and SPAI Short Form (SPAI-SF) [5]. Of these scales, most studies used either the SAS-SV or SAS [5]. Kwon et al. [10] developed the SAS which consists of 33 items and six factors. It was the first questionnaire for the measurement and diagnosis of smartphone addiction, and they showed that the SAS has good validity and reliability [10]. Alternatively, the authors of the SAS also developed the short version of the questionnaire (SAS-SV) which clusters 10 items under one factor [25]. The SAS-SV is intended for easy administration among adolescents, while the SAS is mainly for use among adults [10,25]. The SAS-SV showed comparably good validity and reliability to the longer version of the questionnaire [25]. The SAS-SV has been translated and validated in several languages; Arabic [26], Chinese [27], French [28], German [4], Italian [29], Portuguese [30], Spanish [28] and Turkish [31]. The SAS has been translated and validated in Arabic [26], Japanese [32], Romanian [33] and Thai [34]. In order to facilitate further research on smartphone addiction in Malaysia, Ching et al. [35] translated the SAS into the Malay language (SAS-M) and they showed that the SAS-M is valid and reliable for use among students. The SAS-M has been used in a number of research projects [21,23] in Malaysia to study the effects of smartphone use on students.

However, in both of the SAS-M validation studies [10,35], the researchers used exploratory factor analysis (EFA) to analyze the internal factor structure of the SAS. As compared to EFA, confirmatory factor analysis (CFA) provides stronger evidence to support the validity of the factor structure of a measure [36]. Although CFA provides a better option of analysis, both studies [10,365 were exploratory in nature, which justified the use of EFA. CFA requires specialized software, for example Mplus [37] and *lavaan* R package in R software environment [38,39], while EFA can be easily conducted in many kinds of statistical software. This could have hindered the use of CFA in previous SAS validation studies (with exception of [32,33]). Thus, this study aimed to validate and determine the psychometric properties of the SAS-M by CFA to provide stronger validity and reliability evidence to the earlier EFA study [35].

## 2. Materials and Methods

### 2.1. Study Setting and Participants

A cross-sectional study was conducted among Year 1 to Year 5 undergraduate medical students from School of Medical Sciences, Universiti Sains Malaysia (USM), Kubang Kerian, Kelantan, Malaysia. The data collection was done between October 2016 and December 2016. The sample size was determined for the CFA, which required at least 300 respondents for a scale with seven or less factors with item communality of <0.45 [40]. After taking into consideration a 20% drop-out rate, the target sample size was 375 respondents.

The students were selected by stratified random sampling based on the list obtained from the academic office, with the group size per year of study as the stratification variable. The selected students were then invited to participate in the study. The nature of the study and the confidentiality of their responses was explained. Those who consented to participate in the study signed informed consent forms and were given questionnaire forms to be completed. Each questionnaire form consists of socio-demographic information (age, gender, race and family income), purpose of using smartphones, the SAS-M and the MVIAT. The students returned the completed questionnaire forms to the researchers on the same day.

### 2.2. Instruments

#### 2.2.1. The Malay Version of the Smartphone Addiction Scale (SAS-M)

The English version of the SAS questionnaire [10,35] is a self-administered, 6-point Likert type scale consisting of 33 items that belong to six factors, which are: 1. Cyberspace oriented relationship (7 items): S20–S26; 2. Daily Life Disturbance (5 items): S1–S5; 3. Tolerance (3 items): S31–S33; 4. Overuse (4 items): S27–S30; 5. Positive Anticipation (8 items): S6–S13; and 6. Withdrawal (6 items): S14–S19. Each item has a response scale from 1 (strongly disagree) to 6 (strongly agree). Higher scores indicate a higher degree of smartphone addition [10,35]. The SAS showed good internal consistency reliability (Cronbach’s alpha = 0.967 (total), 0.825 to 0.913 (subscales)) and concurrent validity (*r* = 0.32 to 0.61) [10].

The Malay version of the SAS (SAS-M) was validated among medical students in Malaysia [35]. It showed good internal consistency reliability (Cronbach’s alpha = 0.940 (total), 0.837 to 0.877 (subscales)), test-retest reliability (intraclass correlation (ICC) = 0.85) and concurrent validity (*r* = 0.645) with the Malay version of the Internet Addiction Test (MVIAT). Based on EFA, the number of extracted factors from SAS-M matched the original SAS [35]. As detailed in the study [35], a new factor “Primacy” replaced the “Tolerance” factor from the original SAS, and there were a number of changes to the placement of items under each factor.

In contrast to the original SAS, the SAS-M questionnaire [35] consists of 33 items that belong to six factors, which are: 1. Cyberspace oriented relationship (7 items): S19–S24, S26; 2. Daily Life Disturbance (6 items): S1–S5, S33; 3. Primacy (5 items): S10-S14; 4. Overuse (7 items): S25, S27–S32; 5. Positive Anticipation (4 items): S6–S9; and 6. Withdrawal (4 items): S15–S18.

#### 2.2.2. The Malay Version of the Internet Addiction Test (MVIAT)

The Internet Addiction Test (IAT) is the most commonly used questionnaire to diagnose internet addiction [35]. The questionnaire is a self-administered, 5-point Likert type scale consisting of 20 items, and each item has a response scale from 1 (never) to 5 (always), reflecting the frequency of the symptoms [41]. Higher scores indicate higher degree of pathological use of the internet [41]. The Malay version of the IAT (MVIAT) was validated among a group of medical students in Malaysia; it showed good internal consistency reliability (Cronbach’s alpha = 0.91) and parallel reliability with the IAT (intraclass correlation coefficient (ICC) = 0.88, *p* < 0.001) [41]. The MVIAT [41] showed five extracted factors, which are: 1. Lack of control (8 items): T1, T2, T10, T11, T12, T14, T16, T17; 2. Neglect of Duty (7 items): T3, T6, T8, T9, T18, T19, T20; 3. Problematic Use (2 items): T4, T15; 4. Social relationship disruption (2 items): T5, T13; and 5. Email Primacy (1 item): T7.

### 2.3. Statistical Analysis

The data analysis was performed in R software environment [39]. The CFA was performed using *lavaan* and *semTools* R packages [38,42]. Robust maximum likelihood estimator was used because the data were not normally distributed [36]. The model fit assessment was based on the following fit indices (robust) with their respective cut-off values [36]: *χ*^2^ (*p* > 0.05), comparative fit index (CFI) and Tucker-Lewis fit index (TLI) ≥ 0.95 (good) or ≥0.90 (acceptable), root mean square error of approximation (RMSEA) ≤ 0.08, and standardized root mean square residual (SRMR) ≤ 0.08. Model-to-model comparison was based on Akaike information criterion (AIC) and Bayesian information criterion [36]. The model with the lowest values of AIC and BIC was chosen as the best fitting model for the CFA. Items with factor loadings of >0.3 were considered acceptable [40]. Multicollinearity between the factors was identified when factor-to-factor correlation was *r* > 0.85 [36]. The internal consistency reliability was determined by Raykov’s rho coefficient [43]. Raykov’s rho values of ≥0.7 were considered to reflect good reliability [40].

The concurrent validity was assessed by Pearson’s correlation coefficient (*r*) for the correlations between factor scores of the SAS-M and MVIAT. The strength of correlation was interpreted according to Colton [44]: no or little correlation (0.00–0.25), fair correlation (0.26–0.50), moderate correlation (0.51–0.75), and perfect correlation (0.76–1.00).

### 2.4. Ethical Consideration

The approval for this research was obtained from the Human Research Ethics Committee of Universiti Sains Malaysia (JEPeM Code: USM/JEPeM/16030107). The confidentiality of the data was maintained and only the researchers had access to the data. The permission to conduct the study among the medical students in USM was obtained from the academic office. The permission to use the SAS-M and the MVIAT was obtained from the original authors.

## 3. Results

A total of 323 medical students agreed to participate in the study. The socio-demographic characteristics of the respondents are displayed in Table 1. The mean (SD) age of the students was 21.39 (1.71) years. The majority of them were female (64.4%), of Malay race (53.6%) and those coming from families with a monthly income of more than RM5000 (38.7%).

The CFA of the six-factor model of SAS-M (Model 1) showed that the model did not fit the data. After the inclusion of 12 correlated errors to the model, the CFA showed an acceptable model fit (Model 2). The factors were not multicollinear because all *r* values were less than 0.85, which indicated good discrimination between the factors. The model fit details, factor-to-factor correlation and correlated errors are presented in Table 2.

The factor loadings for all items exceeded the cut-off value of 0.3 (0.320 to 0.875). The reliability for each factor was good (Raykov’s rho = 0.713 to 0.858). The factor loadings and Raykov’s rho values by factor are displayed in Table 3.

The strength and direction of the correlation between the factor scores of the SAS-M and MVIAT are shown in Table 4. The result shows significant correlations between the related SAS-M and MVIAT factors in the range of 0.258 to 0.511, which indicated concurrent validity of the SAS-M.

## 4. Discussion

EFA was used to analyze the internal structure of the original SAS [10] and the Malay version of the SAS [35]. As CFA can provide a stronger evidence in support of the validity of the factor structure [36], this study provided the psychometric properties and model fit assessment of the SAS-M by CFA.

The results of this study showed that the six-factor model of the SAS-M has an acceptable model fit after the inclusion of 12 correlated errors. These correlated errors can be justified on the basis of method effect that reflects additional item covariation, for example between similarly worded items [36]. In the context of the SAS and the SAS-M, the correlation is possible because all items contained the “smartphone” word, and most items contain the “feeling” word. One notable example in this study is between S20 and S21; both item statements start with “Feeling” (SAS) or “Berasa” (SAS-M).

To the authors’ knowledge, this is the first study that reports the CFA of the SAS-M. For the CFA of the SAS, there are a very limited number of studies that used the CFA method [32,33]. The five-factor model is supported mainly by SRMR and RMSEA values that were clearly below the cut-off value of 0.08, while the support from CFI and TLI values were just close enough to the cut-off value of more than 0.90. In contrast, the CFA of the Romanian version of the SAS supported bi-factor model [13], and similar to this study the RMSEA and SRMR values were below 0.08 (0.064 and 0.054 respectively), while the CFI and TLI values were below 0.90 (0.870 and 0.848 respectively). For the Japanese version, they could not confirm the factor structure by CFA and resorted to using EFA which extracted four factors instead of six from the original SAS [32]. In this study, the approach to the CFA was to maintain the six-factor structure found in the earlier EFA [35], to keep all 33 items and to avoid overfitting the model. For these reasons, although it was possible to add more correlated errors to obtain higher CFI and TLI values, the modification to the model was stopped when these values were very close to the cut-off value of 0.90 to qualify as an acceptable model fit. In addition, similar to Vintila et al. [33], the decision to accept the model fit was also based on two other fit indices, which were RMSEA and SRMR. In both studies, the RMSEA and SRMR values were below 0.08 which indicated good model fit.

The factor loadings in each of the factors of the SAS-M showed all items were acceptable in view of the recommended cut-off value of 0.3 [40]. The factor loadings in this study ranged from 0.320 to 0.875. The result is comparable to the factor loadings obtained from the EFA of the SAS-M [35], which ranged from 0.415 to 0.897. The internal consistency reliability for each factor was also good, which ranged from 0.713 to 0.858 (by Raykov’s rho). However, these values were lower than the EFA, which ranged from 0.837 to 0.877 (by Cronbach’s alpha) [35]. This can be attributed to the Raykov’s rho formula, which also takes into account the correlated error covariances in the calculation of the reliability [43].

In this study, most of the factor scores of the SAS-M were correlated to the MVIAT factor scores at fair-to-moderate strength. Similar to the earlier SAS-M validation study [35], there was negligible correlation between the “Positive anticipation” factor score with all MVIAT factor scores. In contrast to the earlier study which examined the correlation between SAS-M factor scores with the total score of the MVIAT [35], this study examined the correlation with each MVIAT factor score. The correlation was low between all factor scores of the SAS-M and “Email primacy” factor score of the MVIAT, and between the “Overuse” factor score of the SAS-M and “Social relationship disruption” factor score of the MVIAT. The poor correlation between the SAS-M and “Email primacy” of the MVIAT could be because email was not the main cause of smartphone addiction. Previous studies indicated that the addiction is better attributed to the social media use of smartphone [12,20]. The poor relationship between “Overuse” of the SAS-M and “Social relationship disruption” of the MVIAT in this study was unexpected because this finding contradicts previous studies which associate problematic smartphone use with poor social interaction [17,18,19]. This could be a population-specific finding (medical students), which warrants a further investigation in future studies.

There are two important limitations of this study that must be highlighted. First, the sample that was used for this study came from a single medical school in Malaysia, which might not represent all medical students in Malaysia. This limitation also affects the earlier study by Ching et al. [35], which also consists of a sample from a single medical school. Thus, it is recommended to cross-validate the SAS-M among medical students from other universities in Malaysia. Second, similar to the study by Ching et al. [35], all participants were medical students. The homogeneity of the sample makes it difficult to generalize the findings to students from other field of studies. In contrast, the validation of the original SAS was done among adult participants from companies and universities [10]. Thus, the validity of the SAS-M could be limited to a medical student population. To address this limitation, it is recommended to cross-validate the questionnaire in other heterogeneous samples in future studies. Despite these limitations, it was decided that in this study medical students would be selected as the sample to reflect closely the population in the EFA study by Ching et al. [35]. By restricting the sample to medical students, this will allow a direct comparison between the factor structure extracted by EFA and the one confirmed by CFA in the selected population.

## 5. Conclusions

This study provided the validity evidence and psychometric properties of the SAS-M by CFA. The results showed that the SAS-M is a valid and reliable self-administered questionnaire to measure the level of smartphone addiction among medical students. The generalizability of this validity evidence can be further supported by conducting confirmatory studies among medical students in other universities in Malaysia. In addition, the use of SAS-M in other populations requires further research in the form of cross-validation studies to explore and confirm its psychometric properties in these populations.

## Figures and Tables

**Table 1 ijerph-17-03820-t001:** Socio-demographic characteristics of the medical students (*N* = 323).

Variable		*N* (%)
Age (Mean [SD])		21.39 (1.71)
Year of Study	Year 1 Year 2 Year 3 Year 4 Year 5	62 (19.2) 59 (18.3) 63 (19.5) 61 (18.9) 78 (24.1)
Gender	Male Female	115 (35.6) 208 (64.4)
Race	Malay Chinese India Others	173 (53.6) 77 (23.8) 64 (19.8) 9 (2.8)
Monthly family income (Ringgit Malaysia; RM)	<RM1000 RM1000–1999 RM2000–2999 RM3000–3999 RM4000–4999 >RM5000	33 (10.2) 49 (15.2) 39 (12.1) 48 (14.9) 28 (8.7) 125 (38.7)

**Table 2 ijerph-17-03820-t002:** Model fit indices and correlated errors of the CFA.

Models	χ² (df), *p*	SRMR	RMSEA (90% CI)	CFI	TLI	AIC	BIC
Model 1	1213.1 (480), < 0.001	0.075	0.074 (0.069, 0.079)	0.834	0.817	31,194.3	31,623.5
Model 2 *	931.3 (468), < 0.001	0.067	0.059 (0.054, 0.065)	0.895	0.882	30,895.2	31,369.7

* Correlated errors in Model 2: S3↔S4 (*r* = 0.410, *p* ≤ 0.001), S14↔S15 (*r* = 0.481, *p* ≤ 0.001), S31↔S32 (*r* = 0.378, *p* ≤ 0.001), S33↔S31 (*r* = 0.336, *p* ≤ 0.001), S21↔S23 (*r* = 0.371, *p* ≤ 0.001), S10↔S13 (*r* = 0.292, *p* ≤ 0.001), S23↔S26 (*r* = 0.353, *p* ≤ 0.001), S21↔S26 (*r* = 0.259, *p* = 0.001), S20↔S21 (*r* = 0.218, *p* = 0.002), S1↔S2 (*r* = 0.274, *p* = 0.002), S14↔S17 (*r* = 0.220, *p* = 0.001), S22↔S30 (*r* = 0.282, *p* = 0.001). Correlation between factors in Model 2: S1↔S2 (*r =* 0.538, *p* ≤0.001), S1↔S3(*r =* 0.762, *p* ≤ 0.001), S1↔S4(*r =* 0.635, *p* ≤ 0.001), S1↔S5 (*r =* 0.422, *p* ≤ 0.001), S1↔S6 (*r =* 0.844, *p* ≤ 0.001), S2↔S3 (*r =* 0.483, *p* ≤ 0.001), S2↔S4 (*r =* 0.694, *p* ≤0.001), S2↔S5 (*r =* 0.292, *p* = 0.004), S2↔S6 (*r =* 0.623, *p* ≤ 0.001), S3↔S4 (*r =* 0.587, *p* ≤ 0.001), S3↔S5 (*r =* 0.676, *p* ≤ 0.001), S3↔S6 *(r =* 0.808, *p* ≤ 0.001), S4↔S5 (*r =* 0.388, *p* ≤ 0.001), S4↔S6 (*r =* 0.697, *p* ≤ 0.001), S5↔S6 (*r =* 0.513, *p* ≤ 0.001). Abbreviations: F = factor; S = SAS item; CFA = confirmatory factor analysis; SRMR = standardized root mean square residual; RMSEA = root mean square error of approximation; CFI = comparative fit index; TLI = Tucker-Lewis fit index; AIC = Akaike information criterion; BIC: Bayesian information criterion.

**Table 3 ijerph-17-03820-t003:** The factor loadings and Raykov’s rho values by factor.

Factor	Question No.	Factor Loading	Raykov’s Rho
F1. Cyberspace oriented relationship	S19 S20 S21 S22 S23 S24 S26	0.592 0.621 0.546 0.791 0.651 0.531 0.562	0.756
F2. Daily life disturbance	S1 S2 S3 S4 S5 S33	0.592 0.637 0.543 0.485 0.654 0.582	0.713
F3. Primacy	S10 S11 S12 S13 S14	0.667 0.783 0.792 0.809 0.735	0.858
F4. Overuse	S25 S27 S28 S29 S30 S31 S32	0.483 0.320 0.585 0.699 0.836 0.786 0.586	0.798
F5. Positive anticipation	S6 S7 S8 S9	0.752 0.875 0.677 0.668	0.831
F6. Withdrawal	S15 S16 S17 S18	0.767 0.743 0.699 0.604	0.800

**Table 4 ijerph-17-03820-t004:** Pearson’s correlation coefficients (*p*-values) between the Malay Version of the Smartphone Addiction Scale (SAS-M) and Malay version of the Internet Addiction Test (MVIAT) factor scores.

	SAS-M					
MVIAT	*Cyberspace oriented Relationship*	*Daily Life Disturbance*	*Primacy*	*Overuse*	*Positive Anticipation*	*Withdrawal*
***Lack Of control***	0.331 (<0.001)	0.437 (<0.001)	0.338 (<0.001)	0.511 (<0.001)	0.211 (<0.001)	0.466 (<0.001)
***Neglect of duty***	0.455 (<0.001)	0.454 (<0.001)	0.320 (<0.001)	0.422 (<0.001)	0.116 (0.220)	0.463 (<0.001)
***Problematic use***	0.400 (<0.001)	0.367 (<0.001)	0.291 (<0.001)	0.310 (<0.001)	0.101 (0.270)	0.393 (<0.001)
***Social*** ***relationship***	0.402 (<0.001)	0.383 (<0.001)	0.258 (<0.001)	0.214 (<0.001)	0.051 (0.360)	0.426 (<0.001)
***Email primacy***	0.243 (<0.001)	0.130 (0.130)	0.097 (0.270)	0.107 (0.270)	0.099 (0.270)	0.158 (0.030)

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
