# Peer review of "Confirmatory Factor Analysis of the Malay Version of the Smartphone Addiction Scale among Medical Students in Malaysia"

_ijerph, 2020, doi:10.3390/ijerph17113820_

Round 1

Reviewer 1 Report

The authors have responded to reviewers' feedback and made necessary changes in a timely manner. The introduction of manuscript is clear and well-written in general. In the mentioned surveys in Introduction, it is recommended to add information about survey sample and response rate, etc. The validity and reliability of instruments in Materials and Methods session are explained clearly. The results are clearly presented. The limitation and implication is addressed. Some minor language issue exists and the references should follow APA style. 

Reviewer 2 Report

The authors have made relevant and sufficient revisions carefully based on the comments/questions I made, which further improved the quality of the manuscript. 

Reviewer 3 Report

Impressive work! I see that there have already been adjustments made to the study, so I don't see any room for improvement there. 

This manuscript is a resubmission of an earlier submission. The following is a list of the peer review reports and author responses from that submission.

Round 1

Reviewer 1 Report

This is an important topic to examine the validity and reliability of Malay version of the Smartphone Addiction Scale (SAS-M) by exploratory factor analysis (EFA) and understand the impact of smartphone addition on student's behaviors. The authors have well explained the research method and data analysis. The recommendations for improving the quality of this manuscript include 1) adding more cited sources to describe the problem, 2) expanding literature review about the existing study of smartphone addiction and SAS and the impact on student's behaviors, 3) improving the explanation of result through connecting the existing literature review and the result of this study, and providing illustration about the impact of smartphone addiction on students' behaviors, and 4) revising the problematic conclusion due to study limitations.

Author Response

Response to Reviewer 1 Comments

Thank you to the reviewer for the constructive comments. We provided the responses for each of the comments below.

Point 1: This is an important topic to examine the validity and reliability of Malay version of the Smartphone Addiction Scale (SAS-M) by exploratory factor analysis (EFA) and understand the impact of smartphone addition on student's behaviors. The authors have well explained the research method and data analysis.

Response 1: Thank you for your acknowledging the importance of the topic. Indeed, in Malaysia there are not many validated questionnaires to address a number of pressing issues in research.

Point 2: The recommendations for improving the quality of this manuscript include 1) adding more cited sources to describe the problem.

Response 2: Thank you for the recommendation. We have provided more explanation about the problem in the Introduction section of the revised manuscript.

Point 3: 2) expanding literature review about the existing study of smartphone addiction and SAS and the impact on student's behaviors.

Response 3: Thank you for the recommendation. We have included more explanation about this issue and its relationship to students’ behaviours in the Introduction section of the revised manuscript.

Point 4: 3) improving the explanation of result through connecting the existing literature review and the result of this study, and providing illustration about the impact of smartphone addiction on students' behaviors.

Response 4: Thank you for your comment. We have expanded the literature review in the Introduction section. As for the connection between the literature and results, we focused our discussion on the psychometric properties of the SAS-M in relation to the SAS-M and the original SAS (EFA). We also relate our findings to the Romanian version, in which they conducted CFA. We have now included the Japanese version to the discussion, in which CFA was conducted for the validation. However, we do not discuss about the EFAs in other translations because this falls out of our scope of matching/confirming the factor structure of the SAS-M, in which each of the translations and cross-cultural adaptations have their own specific situations and limitations. Thus, we focused on the comparison with the other CFAs.

In relation to the results and the impact of smartphone addiction on students’ behavior, we could only partially address this aspect because our study mainly addressed the psychometric properties of the measure (SAS-M) and the relationship with the MVIAT. Within the scope of this study, we could recommend the use of SAS-M among medical students, while acknowledging that further studies will provide more support to its validity. We could also provides insight into the relationship between the SAS-M scores and the MVIAT scores, which we included as follows in the Discussion section:

The poor correlation between the SAS-M and “Email primacy” of the MVIAT could be because email was not the main cause of smartphone addiction. Previous studies indicated that the addiction is better attributed to the social media use of smartphone [20, 12]. The poor relationship between between “Overuse” the SAS-M and “Social relationship disruption” of the MVIAT in this study was unexpected, because this finding contradicts previous studies which associate problematic smartphone use with poor social interaction [17, 18, 19]. This could be a population-specific finding (medical students), which warrants a further investigation in future research.

However, we could not provide specific recommendation about the impact of smartphone addiction from this study per se. This will require a separate study to address this issue, which may utilize the validated SAS-M to measure the level of smartphone addiction. For example, Ithnain et al. (2018) and Nasser et al. (2020) used SAS-M to study the effect of smartphone addiction on undergraduate students. We cited both of these studies in the revised Introduction.

Point 5: 4) revising the problematic conclusion due to study limitations.

Response 5: Thank you for the comment. We realized that we only highlighted the validity of the SAS-M without addressing the limitations in the Conclusion. We added the following sentence to the Conclusion to highlight these limitations:

The generalizability of this validity evidence can be further supported by conducting confirmatory studies among medical students in other universities in Malaysia. In addition, the use of SAS-M in other populations require further research in form of cross-validation studies to explore and confirm its psychometric properties in these populations.

We added the following sentences to address the limitations in the Discussion (last paragraph, Discussion):

Thus, the validity of the SAS-M could be limited to medical students’ population. To address this limitation, it is recommended to cross-validate the questionnaire in other heterogeneous samples in future studies. Despite these limitations, it was decided that in this study, medical students were selected as the sample to reflect closely the population in the EFA study by Ching et al. [36]. By restricting the sample to medical students, this will allow a direct comparison between the factor structure extracted by EFA to the one confirmed by CFA in the selected population.

Reviewer 2 Report

I read this paper with great interests.
One thing I would like to know more is, as the authors suggest, “As compared to EFA, confirmatory factor analysis (CFA) provides a stronger evidence to
54 support the validity of the factor structure of a measure” (line 53), are there any special reasons why CFA has not been conducted in previous studies? Besides, as the authors also pointed out some limitations of their paper at the end of their paper, do they have plans to further improve this study? 

For someone who is not familiar with smartphone addition in Malaysia, it would be very interesting and helpful to have some discussions of the current situations in Malaysia in the introduction part. Is smartphone addition a serious problem there? 

Typo line 81: two “the”s found.

Author Response

Response to Reviewer 2 Comments

Thank you to the reviewer for the constructive comments. We provided the responses for each of the comments below.

Point 1: “As compared to EFA, confirmatory factor analysis (CFA) provides a stronger evidence to 54 support the validity of the factor structure of a measure” (line 53), are there any special reasons why CFA has not been conducted in previous studies?

Response 1: Thank you for your comment. We have now included the following explanation in the text (Introduction section):

Although CFA provides a better option of analysis, both studies [10, 36] were exploratory in nature, which justified the use of EFA. CFA requires specialized software, for example Mplus [38] and lavaan R package in R software environment [39, 40], while EFA can be easily conducted in many statistical software. This could have hindered the use of CFA in previous SAS validation studies (with exception of [34] and [33]).

Point 2: Besides, as the authors also pointed out some limitations of their paper at the end of

their paper, do they have plans to further improve this study?

Response 2: There were two limitations that we highlighted in the manuscript. 1. The use of sample from single school. 2. The homogeneity of the sample i.e. medical students. There are two recommendations that we provided to address these limitations and improve the study.

For limitation (1), this also affected the exploratory study by Ching et al [36]. We recommended in the text as “Thus, it is recommended to cross-validate the SAS-M among medical students from other universities in Malaysia.” for a future study. In our study, we did not have the resources to sample the medical students from other universities.

At the same time, we wanted to replicate the setting of the study by Ching et al [36] so as to provide the confirmatory evidence to the SAS-M factor structure. We added a few points to strengthen the point of including only medical students in this study. We added the following sentences to the text:

Despite these limitations, it was decided that in this study, medical students were selected as the sample to reflect closely the population in the EFA study by Ching et al. [36]. By restricting the sample to medical students, this will allow a direct comparison between the factor structure extracted by EFA to the one confirmed by CFA in the selected population.”

For limitation (2), we originally recommended in the text as “Thus, the validity of the SAS-M could be limited to medical students’ population, and it is recommended to cross-validate the questionnaire in other heterogeneous samples.” for a future study. To make the recommendation clear, we rephrased the sentence as:

Thus, the validity of the SAS-M could be limited to medical students’ population. To address this limitation, it is recommended to cross-validate the questionnaire in other heterogeneous samples in future studies.”.

When we move on from EFA to CFA, there could be a difference in the factor structure if the target populations are different, thus we decided to stick with medical students sample despite its limitation in generalizability for the reason mentioned before (i.e. replicating Ching et al. [36]). In addition, each CFA study requires a very large sample size, of which a minimum of 200 subjects for each of these populations are required. This will require future studies (cross-validation studies) by other researchers as we recommended to provide a bigger picture to the validity of the SAS-M. Again, we did not have the resources to conduct large scale studies.

Point 4: For someone who is not familiar with smartphone addition in Malaysia, it would be very interesting and helpful to have some discussions of the current situations in Malaysia in the introduction part. Is smartphone addition a serious problem there?

Response 4: We have included some background information about smartphone addiction in Malaysia as suggested in the revised manuscript (Introduction section).

Point 4: Typo line 81: two “the”s found.

Response 4: This has been corrected in the revised manuscript. We also made a number of changes to several sentences to improve the sentence composition and grammar as highlighted in the “track-changes”.
